# Lidocaine Enhanced Antitumor Efficacy and Relieved Chemotherapy-Induced Hyperalgesia in Mice with Metastatic Gastric Cancer

**DOI:** 10.3390/ijms26020828

**Published:** 2025-01-19

**Authors:** Peiwen Gao, Fei Peng, Jing Liu, Weiwei Wu, Guoyan Zhao, Congyan Liu, Hangxue Cao, Yuncheng Li, Feng Qiu, Wensheng Zhang

**Affiliations:** 1Department of Anaesthesiology, West China Hospital, Sichuan University, Chengdu 610041, China; 2022324025310@stu.scu.edu.cn (P.G.); lj98_ivy@163.com (J.L.); wuweiwei@scu.edu.cn (W.W.); 2Laboratory of Anaesthesia and Critical Care Medicine, National-Local Joint Engineering Research Centre of Translational Medicine of Anaesthesiology, West China Hospital, Sichuan University, Chengdu 610041, China; pf821@163.com (F.P.); 18328522471@163.com (G.Z.); lcy8697@163.com (C.L.); chxpinocchio@163.com (H.C.); xx17738325251@163.com (Y.L.)

**Keywords:** chemotherapy-induced hyperalgesia, gastric cancer, intraperitoneal suffusion, local anesthetics, peritoneal metastasis, synergistic antitumor

## Abstract

With the widespread use of lidocaine for pain control in cancer therapy, its antitumor activity has attracted considerable attention in recent years. This paper provides a simple strategy of combining lidocaine with chemotherapy drugs for cancer therapy, aiming to relieve chemotherapy-induced pain and achieve stronger antitumor efficacy. However, there is still a lack of substantial pre-clinical evidence for the efficacy and related mechanisms of such combinations, obstructing their potential clinical application. In this study, we propose intraperitoneal chemotherapy (IPC) against gastric cancer (GC) as an ideal scenario to evaluate the efficacy of a lidocaine/paclitaxel combination. Firstly, we used human GC cells MKN-45-luc to investigate the antitumor activity and related mechanisms of the lidocaine/paclitaxel combination in vitro. Then, we used C57BL/6 mice with intraperitoneal drug suffusion to evaluate the efficacy of lidocaine to suppress paclitaxel-induced hyperalgesia and related mechanisms. Lastly, in BALB/c tumor-bearing nude mice we evaluated the synergistic antitumor activity and pain-relieving effect of the lidocaine/paclitaxel combination. Our results showed enhanced antitumor activity for the lidocaine/paclitaxel combination, which induced apoptosis, inhibited migration, and the invasion of GC cells in a synergistic manner. In animal models, the lidocaine/paclitaxel combination effectively inhibited growth and peritoneal metastasis of the tumor, resulting in prolonged survival time. Meanwhile, lidocaine showed considerable anti-inflammatory activity alongside its anesthetic effect, which, in combination, effectively relieved hyperalgesia induced by paclitaxel. These results suggested that intraperitoneal suffusion with lidocaine/paclitaxel could be a pain-free IPC formulation with enhanced antitumor activity, which could provide a promising treatment for GC with peritoneal metastasis.

## 1. Introduction

In the past decades, local anesthetics (LAs) as a category of non-opioid analgesics have been more and more widely used for pain control in cancer treatment. Typical LAs such as bupivacaine, ropivacaine, and lidocaine (Lido) are characterized by a dimethylphenyl and a tertiary amine connected by an amide group. They can block sodium ion channels on nerve cells and cut off the transduction of pain signals in a reversible and non-addictive manner, so they have been conventionally used to relieve incisional pain during tumor resection, including intraperitoneal administration of Lido for many abdominal tumor surgeries [1]. In addition, chemotherapy-induced pain is another kind of symptom that could benefit from LAs [2,3]. For example, intraperitoneal chemotherapy (IPC) could cause abdominal pain or systemic hyperalgesia, likely because the toxic and irritant chemotherapy drugs could damage healthy organs and lead to chemotherapy-induced peripheral neuropathy (CIPN) [4,5]. As a safe LA, allowed for intravenous administration, Lido has shown promising application in patients suffering from CIPN [6,7].

With the wide use of LAs in cancer treatment for pain control, their potential antitumor activity has been noticed in recent years and has attracted more and more attention [8,9,10]. Specifically, Lido, as one of the most frequently used Las, has been extensively investigated for its antitumor activity, with numerous cellular experiments indicating its broad-spectrum antitumor potential [11]. Although the clinical benefit of Lido in cancer treatment is still controversial, some studies have suggested that when used as a supplement for traditional chemotherapy drugs, Lido could substantially enhance antitumor efficacy [12,13,14].

Actually, relieving chemotherapy-induced pain and enhancing antitumor efficacy are two closely related challenges during cancer chemotherapy, which can work in synergism to improve patients’ compliance to the treatment, as well as the overall quality of rehabilitation. For example, IPC, as a major strategy for treating cancers with peritoneal metastasis, is known to cause peritoneal inflammation and abdominal pain, which greatly limits the tolerable dose of chemotherapy drugs and the outcome of IPC [15,16]. On one hand, this raises the demand for sufficient pain control during IPC. On the other hand, combining different antitumor drugs to achieve synergistic antitumor activity has become a conventional strategy to improve the efficacy of chemotherapy [17,18,19,20].

In this study, we used IPC as an ideal scenario to exploit the efficacy of combining Lido and paclitaxel (PTX) to treat gastric cancer (GC) with peritoneal metastasis, focusing on evaluating the antitumor activity and pain-control efficacy (Figure 1). Firstly, we evaluated the synergistic antitumor effect of the Lido/PTX combination in vitro, which included synergistic apoptosis-inducing, anti-migration, and anti-invasion effects on GC cells. Then, we established a mouse IPC model to evaluate the efficacy of Lido in relieving chemotherapy-induced hyperalgesia, which could be attributed to its local anesthetic efficacy to cut off the transduction of acute pain signals, as well as its anti-inflammatory activity to suppress the secretion of chronic pain-inducing inflammatory cytokine from the dorsal root ganglion (DRG). Lastly, we established a mouse GC peritoneal metastasis model to investigate the antitumor and analgesic effects in vivo, in which a large-volume intraperitoneal suffusion with the Lido/PTX combination inhibited the tumor growth and achieved a much longer survival time, meanwhile effectively suppressing hyperalgesia caused by PTX.

## 2. Results

### 2.1. Effect of Lido/PTX Combination on Cell Viability

As shown in Figure 2A, Lido was able to decrease the viability of MKN-45-luc cells in a concentration-dependent manner. When 0.25~4 mM Lido was combined with 0.5 or 1 nM PTX, the combination achieved much higher inhibiting rates than Lido or PTX alone. As shown in Figure 2B, the calculated combination index (CI) values of the two drugs were lower than 1.0 under all tested concentrations, confirming their synergistic antitumor effect. Similarly, the Lido/PTX combination also exhibited a synergistic efficacy to inhibit the proliferation of another GC cell line HGC-27 (Appendix A).

Then, the combination of 1 mM Lido and 1 nM PTX was tested for its ability to induce apoptosis of MKN-45-luc cells. As shown in Figure 2C,D, the cell apoptosis rate in the Lido group and PTX group was 9.43 ± 2.33% and 11.72 ± 2.78%, respectively, indicating the slight apoptosis-inducing effect of the two drugs, as previously reported [21,22]. The Lido/PTX combination exhibited a synergistic effect on inducing apoptosis, with a much higher apoptosis rate of 21.49 ± 3.59%. Additionally, Western blot analysis also revealed the synergistic apoptosis-inducing effect of the Lido/PTX combination, which promoted a higher expression level of the apoptotic protein BAX than all other groups (Figure 2E).

### 2.2. Effect of Lido/PTX Combination on Cell Migration and Invasion

As shown in Figure 3A,B, when the GC cells were treated with 1 mM Lido, the migration distance was greatly inhibited, with a significant difference between the distance of the Lido group and that of the control group (*p* < 0.001). Meanwhile, 1 nM PTX also showed a mild inhibition effect compared with the control group (*p* < 0.05). Moreover, the Lido/PTX combination resulted in a much lower migration rate compared with the Lido, PTX, and control groups, indicating a synergistic anti-migration effect. Consistently, Figure 3C,D showed that the Lido/PTX combination resulted in a much lower invasion rate compared with the Lido, PTX, and control groups (*p* < 0.001). The Western blot image showed that both Lido and PTX could respectively increase the level of a tumor suppressor protein E-cadherin, which could inhibit the epithelial–mesenchymal transition to reduce tumor migration and invasion [23]. The Lido/PTX combination exhibited a much higher level of E-cadherin, indicating a synergistic inhibition effect on cell migration and invasion (Figure 3E).

### 2.3. Lido Relieved PTX-Induced Hyperalgesia in Mouse Model

A C57BL/6 mouse model was used to test the efficacy of Lido to relieve hyperalgesia induced by PTX (Figure 4A). As shown in Figure 4B, the paw withdrawal threshold (PWT) of the PTX group was much lower than the normal saline (NS) and Lido groups after each injection. This hyperalgesia response in the paw has been routinely used as an index to reflect the abdominal pain induced by PTX [24,25]. Furthermore, the PWT value gradually decreased with repeated PTX injections and remained at a low level 2 days after the last injection, reflecting an accumulative effect of PTX-induced pain and a transition from acute pain to chronic pain, as previously reported [26]. However, animals receiving the Lido/PTX combination maintained relatively high PWT values (above 5 g) compared with PTX alone, suggesting that mechanical hyperalgesia caused by PTX was relieved by Lido. Similar results were obtained in the cold hyperalgesia tests, as shown in Figure 4C,D. PTX alone decreased the tail-flick latency upon ice-water stimulus and the paw withdrawal latency upon acetone stimulus, as previously reported [27]. On the contrary, the Lido/PTX group retained a high latency value approximate to the NS and Lido groups. It should be noted that Lido not only inhibited acute mechanical and cold hyperalgesia caused by PTX after each injection, it also consequently inhibited the chronic pain that occurred 2 days after the last drug injection.

As shown in Figure 4E–G, the expression of TRPV1 and c-Fos in DRG also revealed the effect of Lido in preventing PTX-induced chronic pain. In mice receiving PTX, both the mean fluorescence intensity (MFI) of c-Fos signal and the proportion of c-Fos+ cells in all TRPV1+ cells were increased compared with the control and Lido groups, which indicated a sustained transmission of pain signal through the DRG. On the other hand, in the Lido/PTX group, both the MFI of c-Fos signal and the proportion of c-Fos+ cells dropped back to a level similar to the control and Lido groups, suggesting that no pain signal was generated in this group.

To study the role of PTX and Lido in the development of chronic pain by interfering with inflammatory pathways, inflammation cytokine expression in cultured RAW264.7 cells was assayed. As shown in Figure 5A–C, PTX significantly increased the level of TNF-α, IL-1β, and IL-6, while the Lido/PTX combination exhibited a low level of these cytokines similar to the control group. Furthermore, the hematoxylin and eosin (H&E)-stained tissue sections of stomach and intestine were observed to evaluate inflammation caused by different formulations. In animals injected with PTX, there is an accumulation of inflammatory cells in the stomach and intestine, as pointed out by the black arrows in Figure 5D. However, tissue sections from the Lido/PTX group were as normal as those from the NS and Lido groups. All these results suggested that Lido could effectively inhibit inflammation caused by PTX chemotherapy.

Additionally, as an important pathway in pain signal transmission, DRG may also suffer from PTX-induced inflammation and lead to the occurrence of chronic pain. As shown in Figure 5E,F, DRG from mice receiving PTX showed an elevated expression of TNF-α, which might be caused by the accumulation of PTX in DRG through circulation, as previously reported [28]. Meanwhile, in the Lido/PTX group the expression of TNF-α was suppressed to a normal level, similar to the NS group, suggesting the protective effect of Lido against PTX-induced inflammation.

### 2.4. Lido/PTX Combination Inhibited the Development of Peritoneal Metastatic GC

As shown in Figure 6A, the in vivo antitumor activity and safety of Lido/PTX were evaluated in a peritoneal metastatic GC model, as previously reported [29]. As shown in Figure 6B,C, mice treated with NS showed a rapid increase in tumor signal over time, suggesting the rapid growth and spread of seeded tumors in the abdominal cavity. Lido alone only slightly slowed down the increase in tumor signal, while PTX alone showed a moderate inhibition effect on tumor growth. In the Lido/PTX group, the increase in tumor signal was almost completely suppressed, suggesting strong antitumor effects of the drug combination. Moreover, Figure 6D showed a quick decrease in body weight in the NS and Lido groups, while the drop in body weight of the PTX and Lido/PTX groups was relatively slow, which also reflected their tumor-inhibiting effects. As shown in Figure 6E–H, mice treated with Lido/PTX showed the least number and weight of tumors, which also confirmed its strong antitumor efficacy, especially the inhibition of intraperitoneal tumor metastasis. On the other hand, major organs in all groups exhibited similar histological morphology without obvious damage, suggesting the considerable safety of all formulations, including the Lido/PTX combination (Figure 6I).

### 2.5. Lido/PTX Increased Survival Time and Relieved PTX-Induced Hyperalgesia

Furthermore, a study on survival time was carried out to evaluate the synergistic antitumor effects of Lido/PTX, meanwhile PTX-induced hyperalgesia during and after the treatment was also monitored (Figure 7A). The IVIS images (Figure 7B) and luminescence intensity analysis (Figure 7C) reflected successful inhibition of tumor growth by Lido/PTX, which reproduced the results as described in Figure 6. As shown in Figure 7D,E, all mice in the NS and Lido groups died within 30 days. The survival time of the Lido group was 29.4 ± 0.5 days, which did not show a significant difference compared with the NS group (27.2 ± 0.4 days). Although PTX slightly prolonged survival time to 34.2 ± 1.3 days, all mice in this group died within 36 days. In the Lido/PTX group, three out of five mice survived over 50 days, and one mouse in this group survived until the end of the study (60 days) without any sign of tumor growth. As shown in Figure 7E, the Lido/PTX group achieved a much longer average survival time (50 ± 8.6 days). Furthermore, the change in body weight of each mouse in each group was also consistent with the trend of tumor development (Figure 7F), and none of the mice exceeded the humane endpoint (20% body weight loss) during the study.

As shown in Figure 7G, PTX alone decreased the PWT to under 5 g after drug injection, indicating the occurrence of mechanical hyperalgesia. Similarly, Figure 7H,I confirmed the occurrence of cold hyperalgesia caused by PTX, which decreased the tail-flick latency upon ice-water stimulus and the paw withdrawal latency upon acetone stimulus. Moreover, even 3 days after the last drug injection, the mechanical and cold hyperalgesia caused by PTX still existed, reflecting the occurrence of long-lasting chronic pain. Meanwhile, in the Lido/PTX group, the PWT, tail-flick latency, and paw withdrawal latency remained at relatively high levels, suggesting that the mechanical and cold hyperalgesia were effectively suppressed by Lido. It should be noted that in the NS and Lido groups without PTX, the mechanic and cold hyperalgesia also happened after about 2 weeks, which might be attributed to the occurrence of cancer pain with extensive tumor growth. Compared with other groups, the Lido/PTX group exhibited a relatively high and steady threshold throughout the experiment. Obviously, not only Lido in the combination effectively suppressed PTX-induced hyperalgesia, but also the efficient tumor suppression by the synergistic effect of Lido/PTX prevented the occurrence of cancer pain.

## 3. Discussion

As a nascent force in cancer therapy, LAs have been supposed to inhibit tumor growth through multiple potential mechanisms including ion channel blocking, cancer stem cell inhibiting, microRNA regulating, and so on [30,31,32]. Meanwhile, in this study we found that the antitumor activity of Lido could also be attributed to the improved level of E-cadherin, a crucial protein in the epithelial–mesenchymal transition process which inhibits the migration and invasion of cancer cells. Meanwhile, Lido also increased the level of BAX, a pro-apoptotic protein, to induce apoptosis of cancer cells as previously described [22,23]. All these mechanisms are quite different from those of traditional chemotherapy drugs, providing a huge possibility of synergistic antitumor effects by combining LAs with traditional chemotherapy drugs [33,34]. In this study, we found that Lido could significantly increase the antitumor activity of PTX by inducing apoptosis (Figure 2) and inhibiting cell migration and invasion (Figure 3), which could provide a promising combination for the chemotherapy of GC. By using such a combination, strong antitumor activity could be achieved with a much lower concentration of PTX, so that the reduction of side-effects associated with PTX is also expectable.

Although the antitumor activity of Lido is undeniable, its clinical benefit in tumor therapy is still controversial [35]. A possible reason is that in clinical practices, Lido was usually administrated intravenously and may not be effectively targeted to the tumor site. In our study, we supposed IPC as an ideal scenario to exert the antitumor activity of Lido, especially for tumors with peritoneal metastasis. Differently from traditional systemic chemotherapy, IPC made it possible to expose tumor foci spreading in the peritoneal cavity to very high drug concentrations while limiting the toxicity to remote health organs [36]. This is very important for both Lido and PTX considering their systemic toxicity, especially the central nervous system and heart toxicity of Lido when intravenously administrated. For example, in the IPC model in our study, the safe dose of Lido could be as high as close to the median lethal dose of intravenous Lido in mice [37].

Another key point in our IPC protocol is suffusion in the peritoneal cavity with a large volume (750 μL) of drugs rather than a small volume as reported in some previous studies in which the volume of intraperitoneal drug was usually no more than 200 μL [38,39]. In a pilot study, we have used a Lido/PTX combination with the same dose but a much smaller volume (200 μL) to treat nude mice with peritoneal-metastatic GC, and very poor antitumor efficacy was achieved (Appendix A). This is not surprising, since the drug solution with lower volume was not enough to fully fill the abdominal cavity, and could be rapidly absorbed after intraperitoneal injection, leading to limited exposure of tumor foci to the drugs. As shown in Appendix A, 1 h after injection, the mouse that received 200 μL formulation and had little liquid retention in the peritoneal cavity, while the mouse that received 750 μL formulation still kept a moist state in the peritoneal cavity, suggesting the long-term retention of drugs. This large-volume suffusion strategy somehow resembled the clinical practice of intraperitoneal perfusion, which could ensure long-term exposure of tumor foci to the drugs, leading to strong efficacy in inhibiting tumor growth and metastasis.

Except for tumor suppression, another important achievement in our study is the successful relief of chemotherapy-induced hyperalgesia. In traditional clinical practice, effective management of chemotherapy-related pain usually relies on analgesic drugs targeting opioid receptors. Alternatively, Lido and other LAs could suppress the transduction of pain signals by blocking voltage-gated sodium channels in the cell membrane of neurons [40]. This unique mechanism enables Lido to effectively suppress the pain induced by PTX, while keeping free of severe side effects and the problem of addiction associated with opioids.

In our study, we also observed the development of chronic pain in mice receiving PTX. As a possible cause of chronic pain, CIPN has been reported to be related to PTX-induced upregulation of macrophage inflammatory cytokines [41,42]. The ongoing production of inflammatory cytokines contributes to tissue and DNA damage. Previous studies have shown that LAs could have anti-inflammatory potential by affecting inflammatory cells and reducing the secretion of inflammatory mediators [43]. Correspondingly, we found that Lido could exhibit anti-inflammatory effects on the cellular and animal levels (Figure 5), which might be a possible mechanism for the effective suppression of chronic pain as shown in Figure 4 and Figure 7. This finding expanded the mechanism of Lido in relieving chemotherapy-induced hyperalgesia, pushing it to a more promising stage for pain control during chemotherapy.

Lastly, it should be noted that the Lido/PTX combination was a simple mixture of two clinically approved drugs, and the dose of Lido and PTX used in the animal models was based on the safe dosage used in clinic. So the safety of the combination was absolutely expectable, which has also been confirmed by H&E staining of important organs (Figure 6). Furthermore, there might also be an interplay between the antitumor activity and analgesic effect. On one hand, the successful suppression of tumor growth could prevent the development of cancer pain. On the other hand, the relief of pain could in return improve the life quality of the animals, leading to a better outcome of antitumor therapy. Owing to these mechanisms, a much longer overall survival time was achieved in animals receiving the Lido/PTX combination (Figure 7).

However, the current study still has some limitations which could be further improved for even better outcomes. Firstly, although potential antitumor and anti-inflammation mechanisms of Lido have been preliminarily investigated as described above, more detailed and systematic pathways should be further investigated in the future. Secondly, although the large-volume suffusion strategy has partially addressed the problem of the short duration of the formulation, developing slow-releasing formulations with carrier materials may further prolong the duration and enhance the anti-tumor efficacy. Finally, based on the exciting efficacy of the Lido/PTX combination on animal models, further clinical trials could be carried out to validate its efficacy in human patients.

## 4. Materials and Methods

### 4.1. Cell Lines

The luciferase-labeled human GC cell line MKN-45 (MKN-45-luc) was kindly supplied by the State Key Laboratory of Biotherapy, West China Hospital, Sichuan University. The original MKN-45 cell line (Code: 1101HUM-PUMC000229) was purchased from the National Infrastructure of Cell Line Resource (Beijing, China). The RAW264.7 cell line (code: TIB-71, ATCC, Manassas, VA, USA) derived from murine macrophages was kindly supplied by the Cellular Biology Platform, Core Facility of West China Hospital, Sichuan University. The MKN-45-luc (passage number 4) and RAW264.7 cells (passage number 6) were, respectively, cultured in RPMI 1640 and DMEM medium (Hyclone, Logan, UT, USA) supplemented with 10% fetal bovine serum (Gibco, Oakland, CA, USA), 100 U/mL penicillin, and 100 μg/mL streptomycin in a 37 °C humidified incubator containing 5% CO_2_.

### 4.2. In Vitro Cytotoxicity Assay

MKN-45-luc cells were seeded in a 96-well plate at a density of 1 × 10^4^ cells/well and cultured for 24 h. To prove the synergistic antitumor activity of Lido (Sigma-Aldrich, St. Louis, MO, USA) combined with PTX (Meilunbio, Dalian, China), the medium was replaced by fresh medium containing Lido at different concentrations ranging from 0 to 4 mM, with or without adding 0.5 or 1 nM PTX. The cells were incubated with drugs for 48 h and cell viability was tested using the Cell Counting Kit-8 (APExBIO, Houston, TX, USA) following the manufacturer’s instructions. The control group was treated with plain culture medium. The optical density (OD) values were detected at 490 nm by using a BioTek Eon microplate spectrophotometer (BioTek Instruments Inc., Winooski, VT, USA). Cell viability was calculated as follows:Cell viability (%) = (ODtest − ODblank)/(ODcontrol − ODblank) × 100%

For each formulation, an averaged viability was calculated as the means of three separate experiments. The combination index (CI) value of each different formulation was calculated using CompuSyn software (version 1.0, ComboSyn Inc, Paramus, NJ, USA), and synergism was defined when CI < 1.

### 4.3. Apoptosis Assay

Briefly, MKN-45-luc cells were seeded in 6-well plates at a density of 5 × 10^5^ cells/well, incubated overnight, and treated with fresh medium (control), medium containing 1 mM Lido, 1 nM PTX, or 1 mM Lido plus 1 nM PTX (Lido/PTX). After incubation for another 4 h, cells were collected by trypsinization and stained with Annexin V-PI (Thermo Fisher Scientific, Waltham, MA, USA). The samples were then analyzed using a CytoFLEX flow cytometer (Beckman-Coulter, Brea, CA, USA) to determine the apoptosis ratio.

### 4.4. Scratch Assay

MKN-45-luc cells were seeded in 6-well plates at a density of 5 × 10^5^ cells/well and incubated overnight. When the confluence reached approximately 90%, the cells were scratched with a pipette tip to create a gap on the bottom of the well. Then, the cells were treated with 1 mM Lido, 1 nM PTX, or Lido/PTX containing 1 mM Lido and 1 nM PTX. An untreated group was set as the control. The cells were incubated with different formulations for 24 h. Then, for each well, three random fields at the scratch site were imaged with an OBSERVER D1/AX10 cam HRC inverted microscope (Zeiss, Baden-Württemberg, Germany). The gap area at 0 h and 24 h was measured with ImageJ software (version 1.54d, National Institutes of Health, Bethesda, MD, USA) and the relative migration rate was calculated as follows:Relative migration (%) = (GAT0 − GAT24)/(GAU0 − GAU24) × 100%

GAT0 and GAT24: the gap area of treated cells at 0 h and 24 h. GAU0 and GAU24: the gap area of untreated control cells at 0 h and 24 h.

### 4.5. Cell Invasion Assay

Transwell inserts (Corning, NY, USA) pre-coated with Matrigel (Corning, NY, USA) were used for the cell migration assay. A total number of 1 × 10^4^ MKN-45-luc cells in 200 µL of serum-free medium containing 1 mM Lido, 1 nM PTX, or 1 mM Lido plus 1 nM PTX (Lido/PTX) were added into the upper chamber, while 650 µL of medium with 10% FBS containing corresponding drugs was added into the lower chamber. Medium without drug was used as the untreated control group. After incubation for 24 h, cells attached on the outside bottom of the inserts were fixed with 4% paraformaldehyde, stained with 0.1% crystal violet, and imaged with an inverted microscope. For each sample, at least five random areas on the bottom were imaged and cell numbers were counted. The relative invasion rate was calculated as follows:Relative invasion (%) = NT/NU × 100%

NT: the average number of cells treated with different drugs. NU: the average number of untreated cells.

### 4.6. Western Blotting

MKN-45-luc cells were seeded in 6-well plates, incubated and pretreated with different formulations as described above in the apoptosis assay. Then, the cells were lysed on ice for 10 min with radioimmunoprecipitation assay buffer containing a protease and phosphatase inhibitor cocktail (MedChemExpress, Shanghai, China). Proteins extracted from cell lysates were electrophoresed in a 4–20% gradient SDS-PAGE gel (Beyotime, Haimen, China), and transferred onto a nitrocellulose membrane. The membranes were incubated, respectively, with primary antibodies against E-cadherin (Cat No. 20874-1-AP; 1:20,000), BAX (Cat No. 60267-1-Ig; 1:5000), and GAPDH (Cat No. 10494-1-AP; 1:5000) (Proteintec, San Diego, CA, USA) at 4 °C overnight, then with horseradish peroxidase-conjugated secondary antibodies at 25 °C for 3 h. An enhanced chemiluminescence detection system (Epizyme Biotech, Shanghai, China) was used to visualize immunoreactive bands.

### 4.7. Cytokine Measurement

The RAW246.7 cells at a density of 5 × 10^4^ cells/well in 96-well plates were pretreated with fresh medium (control) or medium containing 1 mM Lido, 10 μM PTX, or Lido/PTX containing 1 mM Lido and 10 μM PTX for 24 h. The expression levels of TNF-α, IL-1β, and IL-6 were determined by mouse ELISA kits (Invitrogen, ThermoFisher Scientific Inc., Waltham, MA, USA) following the manufacturer’s instruction. The detection range of the TNF-α kit (Cat No. BMS607-3TEN) is 8–1000 pg/mL and the detection range of the IL-1β kit (Cat No. BMS6002-2TEN) and IL-6 kit (Cat No. BMS603-2) is 7.8–500 pg/mL. The values were read by a BioTek Eon microplate spectrophotometer (BioTek Instruments Inc., Winooski, VT, USA) and all data were normalized to the control group.

### 4.8. Animals

Female BALB/c nude mice and male C57BL/6 mice (6–8 weeks) were purchased from Vital River Laboratory Animal Technology Co., Ltd. (Beijing, China). Animals were housed in an equipped animal facility with the temperature set at 20–25 °C and humidity set at 50 ± 5% under the same dark/light cycle (12 h/12 h). The animal study protocols were approved by the West China Hospital (Sichuan University, Chengdu, China) Animal Ethics Committee (approval number: 20230515001) and all experiments were conducted in strict accordance with the NIH Guide for Care and Use of Laboratory Animals.

### 4.9. Test of PTX-Induced Hyperalgesia

C57BL/6 mice were randomly divided into 4 groups (*n* = 5) to receive different formulations including NS, Lido, PTX, and Lido/PTX. The dosages of Lido and PTX in different formulations were 30 mg/kg and 5 mg/kg, respectively. For each injection, 750 μL of formulation was slowly injected into the abdomen cavity, and then the abdomen was gently rubbed to ensure the spread of drugs. Each mouse received 4 injections with an interval of 2 days. As previously reported, hyperalgesia induced by PTX could be observed as mechanical and cold hyperalgesia, which were characterized by obvious avoidance behavior in response to mechanical and cold stimuli, respectively [44]. At time points of 1 h after each injection, as well as 24 h and 48 h after the last injection, the mechanical and cold hyperalgesia of mice in different groups were tested as follows.

Briefly, a BIO-EVF4 electronic von Frey filament (Bioseb, Vitrolles, France) was used to quantify mechanical hyperalgesia. The mice were habituated inside a small plastic chamber (5 cm wide × 8 cm long × 5 cm high) with a floor of soft wire mesh. The filament was applied to the plantar surface of the hind paw of the mouse with gradually increasing pinprick pressure until a withdrawal response was observed. The pressure value inducing the withdrawal response was recorded as the withdrawal threshold. A cut-off value was set at 10 g. Cold hyperalgesia of the tail was measured by immersing the tip of the tail in ice water and recording the latency to a withdrawal reflex. A cut-off latency was set at 20 s. Alternatively, cold hyperalgesia of the hind paw was assessed using a modified acetone method. The mouse was grasped with the trunk remaining upright, while the limbs could move freely. Using the tip of a syringe, an acetone-soaked cotton ball was applied to the plantar surface of the hind paw and the latency to a withdrawal reflex was recorded. For each mouse in each test, a baseline was measured before drug injection, and the animal was monitored for mechanical and cold hyperalgesia at preset time points. Mechanical hyperalgesia was defined as a decrease in withdrawal threshold as compared with the baseline. Cold hyperalgesia was defined as a decrease in tail-flick or paw withdrawal latency as compared with the baseline. The animals were anesthetized with sevoflurane and immediately decapitated 2 days after the last drug injection, and DRG of lumbar segments L4/5, stomach, and intestine were collected for following studies.

### 4.10. Immunofluorescence Assay of DRG

DRG samples collected in the pain test were post-fixed with 4% paraformaldehyde, dehydrated, embedded in paraffin, and then sectioned into slices with 3 μm thickness. TRPV1 and c-Fos in DRG were stained using the TRPV1 kit (DF10320, Affinity Biosciences, Melbourne, Australia) and the c-Fos kit (GB12096, Servicebio, Wuhan, China) following the manufacturer’s instructions. Meanwhile, TFN-α expressed in DRG was stained using the TNF-α kit (Affinity, AF7014) following the manufacturer’s instructions. Images were acquired using fluorescence microscopy (ECLIPSE Ti2, Tokyo, Japan). The number of positive neuron cells and MFI were calculated using Image J (version 1.53t, National Institute of Health, MD, USA).

### 4.11. Anticancer and Analgesic Effect on GC Peritoneal Metastasis Model

Nude mice were intraperitoneally injected with 2 × 10^6^ MKN-45-luc cells suspended in 200 μL of PBS. After 5 days, the tumor burden was measured by in vivo bioluminescence imaging using the In Vivo Imaging System (IVScope 8200, Clinx Science Instruments Co., Ltd., Shanghai, China) [45]. The animals were randomly distributed into 4 groups (*n* = 5) with similar average tumor burdens, as estimated by the bioluminescence, and 750 μL of different formulations were injected into the abdomen cavity once every 2–3 days for 6 times within 2 weeks. The formulations included NS, Lido, PTX, and Lido/PTX, and the dosages of Lido and PTX were 30 mg/kg and 5 mg/kg, respectively. The tumor burden was monitored by bioluminescence, as described above. The animals were anesthetized with sevoflurane and immediately decapitated on day 27, and the tumor nodules in each mouse were collected for photographing, weighing, and counting. Organs including heart, liver, spleen, lung, and kidney were also collected.

Alternatively, the survival time of mice receiving the same tumor inoculation and drug injection as described above was assessed (*n* = 5). For mice receiving different formulations, the survival rate until 60 days was recorded and the average survival time of each group was calculated, meanwhile the development of tumor burden was monitored by in vivo bioluminescence imaging in the first 4 weeks, as described above. At the same time, the mechanical and cold hyperalgesia of the nude mice in the survival experiment were also tested at different time points including 1 h after the first, third, and fifth injection, as well as 72 h after the last injection. The methods for the pain test were the same as described in the C57BL/6 mouse model, except that the cut-off latency of cold hyperalgesia for nude mice was set as 10 s.

### 4.12. Histology Analysis

To evaluate the toxicity and safety of different formulations, organs collected from C57BL/6 mice in the pain test (stomach and intestine) and nude mice in the antitumor experiment (heart, liver, spleen, lung, kidney) were fixed with 4% paraformaldehyde solution, conventionally dehydrated, and embedded in paraffin. Then, tissue sections with a thickness of 4 μm were stained with H&E and imaged using a BA210 digital microscope (Motic China Group Co., Ltd., Shenzhen, China).

### 4.13. Statistical Analysis

All results are presented as mean with SD. The normality of distributions was determined by the D’agostino–Pearson test. Experimental results were analyzed with unpaired two-tailed Student’s *t*-test (for normal distribution) or two-tailed Mann–Whitney U test (for non-normal distribution) for two independent groups. A one-way analysis of variance was used for more than two groups, followed by the Holm–Sidak multiple comparisons test (used when all groups were mutually compared) or Dunnett’s test (used for comparing the intervened groups with a control group). Differences with *p* < 0.05, *p* < 0.01, and *p* < 0.001, were considered statistically significant and were labeled with *, **, and ***, respectively. Statistical analysis was performed using GraphPad Prism 8.0 (GraphPad Software Inc., San Diego, CA, USA).

## 5. Conclusions

To sum up, our study introduced a Lido/PTX combination with strong antitumor activity and a sufficient analgesic effect. The strong antitumor activity was achieved by two mechanisms: the synergistic effect of Lido and PTX, and the large-volume intraperitoneal suffusion. Effective pain control was also exhibited in two aspects: effective suppression of acute pain by Lido as a local anesthetic, and inhibition of chronic pain by the anti-inflammatory effect of Lido. Based on the combination of these well-established mechanisms, as well as the fact that both Lido and PTX have been conventionally used in clinical settings, this Lido/PTX combination could be quickly evaluated in clinical practice and deliver a promising IPC strategy for patients suffering from GC with peritoneal metastasis.

## Figures and Tables

**Figure 1 ijms-26-00828-f001:**
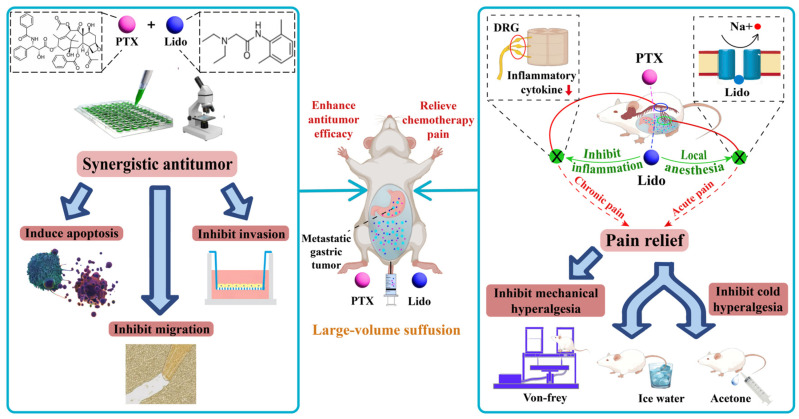
Schematic illustration for investigating the synergistic antitumor and pain-relieving efficacy of the Lido/PTX combination.

**Figure 2 ijms-26-00828-f002:**
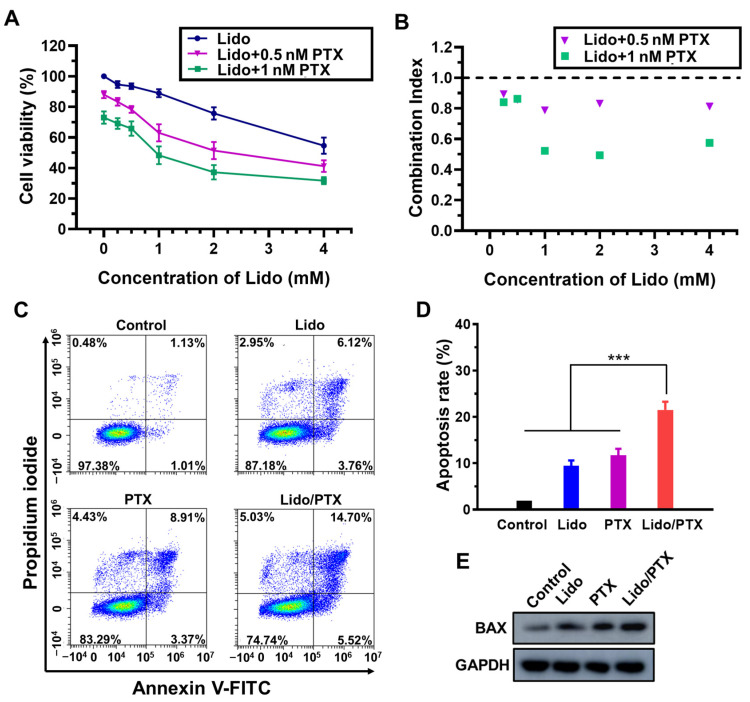
Synergistic effect of Lido and PTX on cell viability and apoptosis of MKN-45-luc cells. (**A**) Viability of cells treated with Lido and PTX at different concentrations (*n* = 3). (**B**) CI values of Lido and PTX at different concentrations. (**C**) Representative flow cytometry images. (**D**) Apoptosis rates (*n* = 4). (**E**) Western blot analysis of apoptotic marker proteins in the control, Lido, PTX, and Lido/PTX groups. GAPDH was used as an internal control. Data are presented as the mean ± standard deviation (SD) and statistical significance was assessed by one-way ANOVA. *** *p* < 0.001.

**Figure 3 ijms-26-00828-f003:**
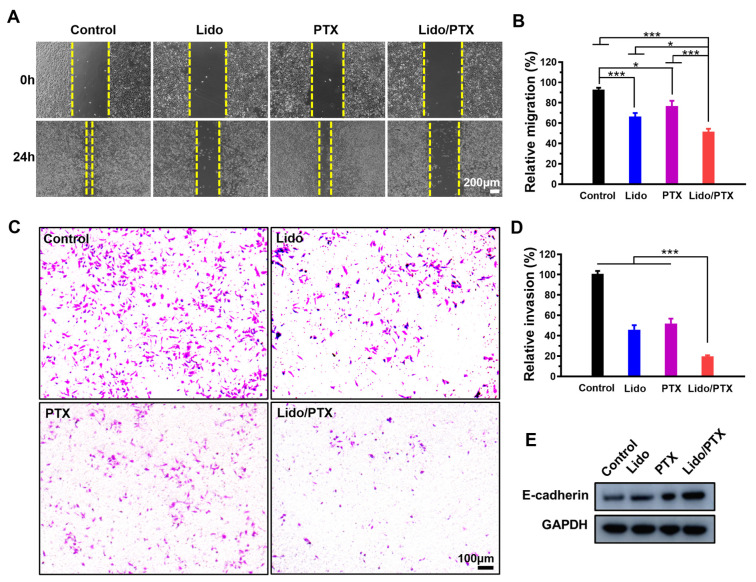
Synergistic effect of Lido and PTX on migration and invasion of MKN-45-luc cells. (**A**) Representative images of cell migration by scratch assay. (**B**) Relative migration rates (*n* = 4). (**C**) Representative images of cell invasion. (**D**) Relative invasion rates (*n* = 4). (**E**) Western blot analysis of E-cadherin in the control, Lido, PTX, and Lido/PTX groups. Data are presented as the mean ± SD and statistical significance was assessed by one-way ANOVA. * *p* < 0.05, *** *p* < 0.001.

**Figure 4 ijms-26-00828-f004:**
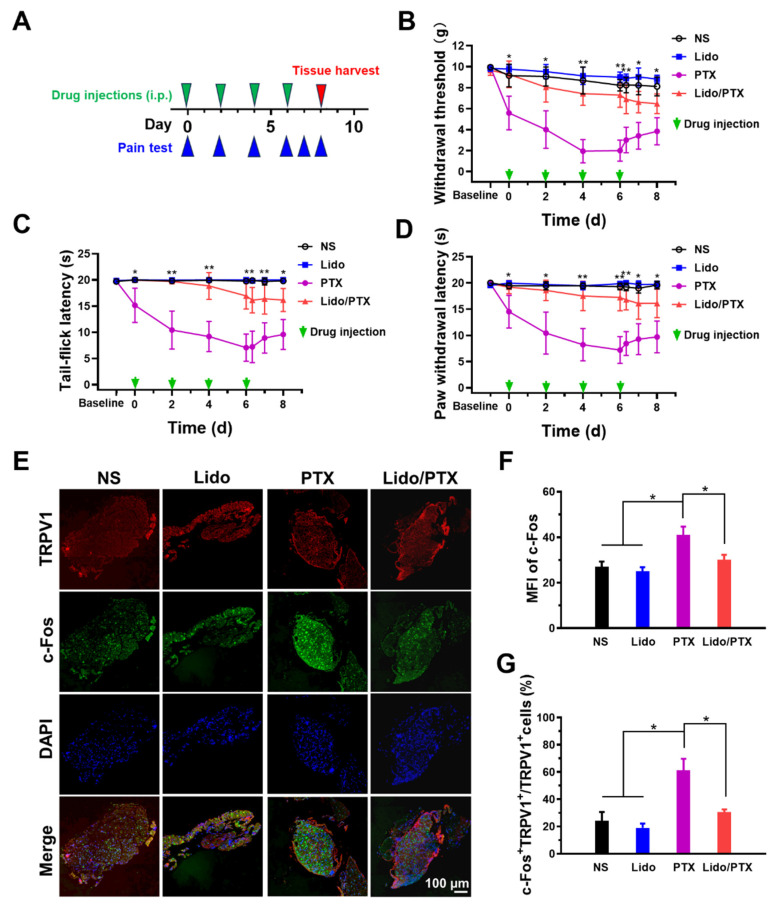
Analgesic effect of Lido on PTX-induced hyperalgesia. (**A**) Schematic diagram of the model establishment and treatment schedule (*n* = 5). (**B**) The PWT in response to mechanical stimulus. (**C**) The tail-flick latency in response to ice-water stimulus. (**D**) The paw withdrawal latency to acetone stimulus. (**E**) Representative immunofluorescence images of c-Fos+ and TRPV1+ cells in DRG (*n* = 3). (**F**) Statistic analysis of MFI of c-Fos signal. (**G**) Statistic analysis of the portion of c-Fos+ cells in all TRPV1+ cells. Data are presented as the mean ± SD and statistical significance was assessed by one-way ANOVA. * *p* < 0.05, ** *p* < 0.01.

**Figure 5 ijms-26-00828-f005:**
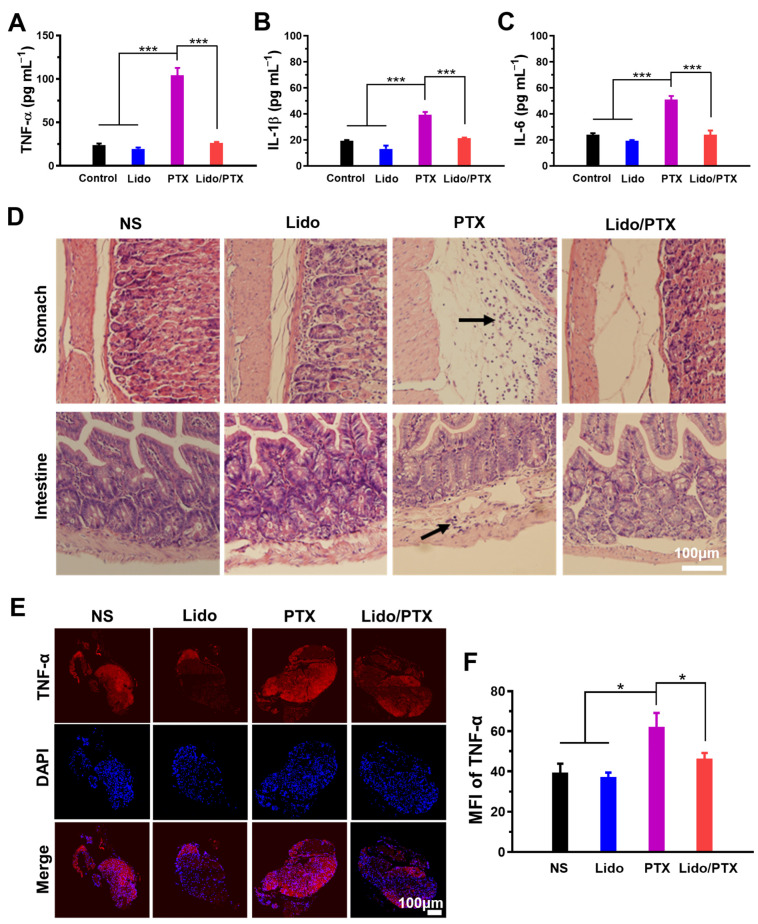
Anti-inflammatory effect of Lido in the Lido/PTX combination. (**A**) TNF-α, (**B**) IL-1β, and (**C**) IL-6 secreted from macrophages treated for 24 h by different groups (*n* = 3). (**D**) Representative images of H&E-stained sections of tissue from different groups (*n* = 3). Accumulation of inflammatory cells was pointed out by the black arrows. (**E**) Representative immunofluorescence images of TNF-α expressed in DRG (*n* = 3). (**F**) Statistic analysis of MFI of TNF-α signal (*n* = 3). Data are presented as the mean ± SD and statistical significance was assessed by one-way ANOVA. * *p* < 0.05, *** *p* < 0.001.

**Figure 6 ijms-26-00828-f006:**
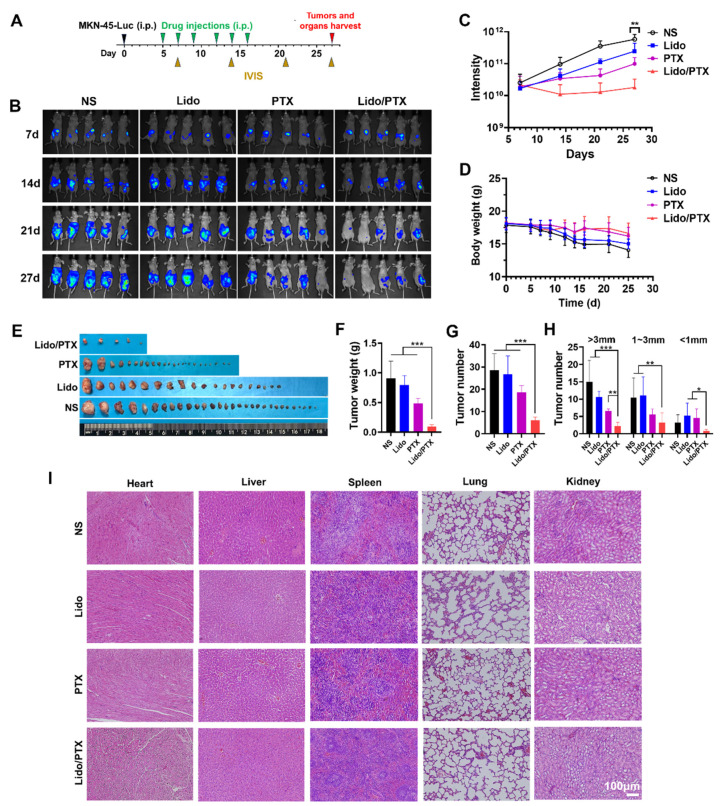
Lido/PTX combination inhibited tumor growth in a peritoneal metastatic GC model. (**A**) Schematic diagram of the model establishment and treatment schedule (*n* = 5). (**B**) Luminescence images obtained through In Vivo Imaging System (IVIS) imaging weekly. (**C**) Quantification of luminescence measurements. (**D**) Body weight of mice. (**E**) Representative photographs of peri-gastric mesenteric tumor nodules on day 27 after different treatments. (**F**) Weight of tumor nodules. (**G**) Total number of tumor nodules. (**H**) Number of tumor nodules in different tumor volume ranges. (**I**) Representative images of H&E-stained sections of tissue from different groups. Data are presented as the mean ± SD and statistical significance was assessed by one-way ANOVA. * *p* < 0.05, ** *p* < 0.01, *** *p* < 0.001 between Lido/PTX and PTX.

**Figure 7 ijms-26-00828-f007:**
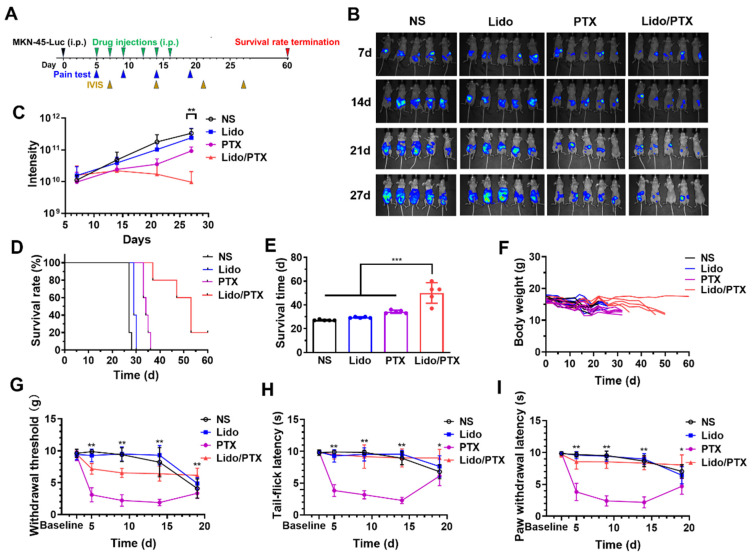
Lido/PTX combination group enhanced survival time and inhibited PTX-induced hyperalgesia in a peritoneal metastatic GC model. (**A**) Schematic diagram of the model establishment and treatment schedule (*n* = 5). (**B**) Luminescence images obtained through IVIS imaging weekly. (**C**) Quantification of luminescence measurements. (**D**) Survival rate, (**E**) survival time, and (**F**) body weight of mice. (**G**) The paw withdrawal threshold by mechanical stimuli. (**H**) The tail-flick latency to ice-water. (**I**) The paw withdrawal latency to acetone. Data are presented as the mean ± SD and statistical significance was assessed by one-way ANOVA. * *p* < 0.05, ** *p* < 0.01, *** *p* < 0.001 between Lido/PTX and PTX.

## Data Availability

The data generated during the current study are available from the corresponding author on reasonable request.

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
