# Peer review of "Lidocaine Enhanced Antitumor Efficacy and Relieved Chemotherapy-Induced Hyperalgesia in Mice with Metastatic Gastric Cancer"

_ijms, 2025, doi:10.3390/ijms26020828_

Round 1
Reviewer 1 Report
Comments and Suggestions for Authors
The article " Lidocaine enhanced antitumor efficacy and relieved chemo- 2 therapy-induced hyperalgesia in mice with metastatic gastric 3 cancer" by Gao and colleagues, aim to present the use of Lido and Paclitaxel that may have antitumor effects.
Introduction
This chapter is well described and the schematic figure (Figure 1) is increasing its readability and helps the readers to understand the aims and scope of the paper.
Results
2.1. It is clear that Lido+PTX have a dose dependent effect, inducing apoptosis in tumor cells, with the most effective proapoptotic effect attributed to Lidp+PTX.
Observation: I recommend using nM instead of micromolar. It would be better for readers, as it is clear that the PTX was added in a nM dose.
The invasion and migration were inhibited by combination of PTX and Lido, results that are well presented in figure 3.
2.3. The in vivo experiements are well conducted, following the general regulations and limited the use of animal to the precise number that was enough for validating the results.
Lido and PTX modulated the inflammatory pathway, as depicted in figure 5, where Lido lowered proinflammatory molecules, both alone and in combination with PTX.
2.4. and 2.5. The results underline the efficacy of Lido in combination with PTX, all tumors were inhibited and the results showed a better response of PTX combined with Lido, than PTX alone.
Discussion section
The authors discussed al aspects needed to sustain the results. This chapter is well structured and written and I have no comments here
Materials and methods
Cell lines-please provide a code for the MKN-45 cell line. The MKN-45 Luc2 cells were generated in-house, i guess, so the original cell line should have a code/cat.no. Same for RAW264.7 cells.
4.2. Please check the density, 10^ 4 should be written.
4.3. Check the density 10^5
4.4. Add the initial seeding density.
4.5. Check cell density 10^4
4.6. How did you prepared the cells? Were they seeded in 6 well plates or in flasks, or petri dish?
4.7. Check seeding density 10^4. Also, add cat no. for each ELISA kit.
4.11. Check the density 10^6
Conclusions
The conclusions are sustained by the presented data. No other comments.
Original blots are looking good and supplemental data are ok.
No issues detected within the references and no plagiarism was detected.
I recommend some revisions for this manuscript before being accepted for publication.
Congratulations for your work!
Author Response
Comments 1: I recommend using nM instead of micromolar. It would be better for readers, as it is clear that the PTX was added in a nM dose. |
Response 1: Thank you for pointing this out. We agree with this comment. Therefore, we have used nM instead of micromolar for dose of PTX in main text and Figure 2A, 2B according to the comment. [page 3, line 85; page 4, line 98, 109; page 14, line 347; page 15, line 360, 368, 380.]
|
Comments 2: Cell lines-please provide a code for the MKN-45 cell line. The MKN-45 Luc2 cells were generated in-house, i guess, so the original cell line should have a code/cat.no. Same for RAW264.7 cells. |
Response 2: Agree. We have, accordingly, provided the code for MKN-45 and RAW264.7 cells in section 4.1 (Cell lines) according to the comment. [page 14, line 333-336.]
|
Comments 3: 4.2. Please check the density, 10^ 4 should be written. |
Response 3: Agree. We have, accordingly, corrected the mistake according to the comment. [page 14, line 343.]
|
Comments 4: 4.3. Check the density 10^5 . |
Response 4: Agree. We have, accordingly, corrected the mistake according to the comment. [page 15, line 358.]
|
Comments 5: 4.4. Add the initial seeding density. |
Response 5: Agree. We have, accordingly, added the initial seeding density in section 4.4 according to the comment. [page 15, line 365-366.]
|
Comments 6: 4.5. Check cell density 10^4 |
Response 6: Agree. We have, accordingly, corrected the mistake according to the comment. [page 15, line 379.]
|
Comments 7: 4.6. How did you prepared the cells? Were they seeded in 6 well plates or in flasks, or petri dish? |
Response 7: The cells were seeded in 6 well plates and the corresponding method of cell preparation has been added to section 4.6 according to the comment.[page 15, line 392.]
|
Comments 8: 4.7. Check seeding density 10^4. Also, add cat no. for each ELISA kit. |
Response 8: Agree. We have, accordingly, corrected the mistake and added cat no. for each ELISA kit in section 4.7 according to the comment. [page 16, line 404 and 408-410.]
|
Comments 9: 4.11. Check the density 10^6 |
Response 9: Agree. We have, accordingly, corrected the mistake according to the comment. [page 17, line 461.]
|
4. Response to Comments on the Quality of English Language |
Point 1: The quality of English does not limit my understanding of the research. |
Response 1: Thank you for the positive comment. Nevertheless, we have checked and revised the manuscript carefully to further improve the quality of English language. |

Reviewer 2 Report
Comments and Suggestions for Authors
Comments
The manuscript with entitled “ Lidocaine enhanced antitumor efficacy and relieved chemotherapy- induced hyperalgesia in mice with metastatic gastric 3 cancer”. The article is related to possible clinical treatments.. However.
1. In the introduction, the author should introduce the chemical structure and molecular formula of Lidocaine.
2. What are the characteristic signs of the model of chemotherapy-induced hyperalgesia? How does the author determine the success of the model of chemotherapy-induced hyperalgesia? The author should have clear indicators of successful model establishment. Recommendation: The author to provide additional clarification.
3. Are Lidocaine clinical available? If it is used clinical available, what is the dose used? What is the relationship between the dose of clinical and the dose used in this study? What is base of the dose of Lidocaine used in this study? Recommendation: The author to provide additional clarification.
4. In the discussion section, the author should focus on the target of action for Lidocaine. How and why does play this role. Recommendation: The author to provide additional clarification.
5. What are the specific limitations of the article? Recommendation: The author to provide additional clarification.
Comments on the Quality of English LanguageThere are many mistakes in grammar and words in the article. Especially in the methods section. Recommendation: Check the spelling and grammar of the text.
Author Response
Comments 1: 1.In the introduction, the author should introduce the chemical structure and molecular formula of Lidocaine. |
Response 1: Thank you for pointing this out. We have added a description on the general chemical structure of local anesthetics including lidocaine in the introduction. The exact chemical structures of lidocaine and PTX were added in Figure 1 for better illustration. With the detailed chemical structure provided, we prefer not to add the molecular formula, since it wouldn’t give any more information. [page 1-2, line 38-42.]
|
Comments 2: What are the characteristic signs of the model of chemotherapy-induced hyperalgesia? How does the author determine the success of the model of chemotherapy-induced hyperalgesia? The author should have clear indicators of successful model establishment. Recommendation: The author to provide additional clarification. |
Response 2: Agree. We have revised section 4.9 “Test of PTX-induced hyperalgesia” based on your recommendation, adding a description of the characteristics of the chemotherapy-induced hyperalgesia model. The detailed criteria for determining mechanical and hyperalgesia were added as requested (page 16-17, line 443-448). Actually, PTX-induced hyperalgesia in mice has been a conventional and reliable method. In our study, every mouse in the PTX group developed hyperalgesia.
|
Comments 3: Are Lidocaine clinical available? If it is used clinical available, what is the dose used? What is the relationship between the dose of clinical and the dose used in this study? What is base of the dose of Lidocaine used in this study? Recommendation: The author to provide additional clarification. |
Response 3: Lidocaine is clinical available and we have added the description as Response 1. The selection of lidocaine dosage is described as below: For adults, a 2% lidocaine solution is commonly used in clinic, with a dose of 200 mg (intraperitoneal injection), which has been proven effective for postoperative pain relief [Clin J Pain, 2010, 26, 121-127.]. According to the species dose conversion ratio between humans and mice, approximately 1:12, the intraperitoneal dose for humans is about 3-4 mg/kg when the body weight is between 50-70 kg. Therefore, the dose converted for mice would be approximately 30-50 mg/kg. Correspondingly, the dose used in the animal models of this study was 30 mg/kg. In the original manuscript we have indicated that “the dose of Lido and PTX used on the animal models was based on the safe dosage used in clinic. So that the safety of the combination was absolutely expectable” (page 14, line 312-314).
|
Comments 4: In the discussion section, the author should focus on the target of action for Lidocaine. How and why does play this role. Recommendation: The author to provide additional clarification. |
Response 4: Agree. We have, accordingly, added the additional clarification of antitumor and anti-inflammation mechanisms for Lidocaine in Discussion. [page 13, line 254-259 and page 14, line 302-305.]
|
Comments 5: What are the specific limitations of the article? Recommendation: The author to provide additional clarification. |
Response 5: Agree. We have, accordingly, added the limitations in Discussion. [page 14, line 321-329.]
|
4. Response to Comments on the Quality of English Language |
Point 1: There are many mistakes in grammar and words in the article. Especially in the methods section. Recommendation: Check the spelling and grammar of the text. |
Response 1: We have checked and revised the manuscript carefully to improve the quality of English language. |

Round 2
Reviewer 2 Report
Comments and Suggestions for Authors
None
Author Response
Comments: None.
Respond: Thank you very much for taking the time to review this manuscript and providing valuable suggestion for the revision before.